# High Rates of COVID-19 Vaccine Hesitancy and Its Association with Conspiracy Beliefs: A Study in Jordan and Kuwait among Other Arab Countries

**DOI:** 10.3390/vaccines9010042

**Published:** 2021-01-12

**Authors:** Malik Sallam, Deema Dababseh, Huda Eid, Kholoud Al-Mahzoum, Ayat Al-Haidar, Duaa Taim, Alaa Yaseen, Nidaa A. Ababneh, Faris G. Bakri, Azmi Mahafzah

**Affiliations:** 1Department of Pathology, Microbiology and Forensic Medicine, School of Medicine, The University of Jordan, Amman 11942, Jordan; mahafzaa@ju.edu.jo; 2Department of Clinical Laboratories and Forensic Medicine, Jordan University Hospital, Amman 11942, Jordan; 3Department of Translational Medicine, Faculty of Medicine, Lund University, 22184 Malmö, Sweden; 4Department of Dentistry, Jordan University Hospital, Amman 11942, Jordan; deemahameddababseh@gmail.com; 5School of Dentistry, The University of Jordan, Amman 11942, Jordan; hda0175066@ju.edu.jo (H.E.); aya0160237@ju.edu.jo (A.A.-H.); duaa.taim@yahoo.com (D.T.); 6School of Medicine, The University of Jordan, Amman 11942, Jordan; Klo0180363@ju.edu.jo; 7School of Science, The University of Jordan, Amman 11942, Jordan; alaa.yaseen@ju.edu.jo; 8Cell Therapy Center (CTC), The University of Jordan, Amman 11942, Jordan; n.ababneh@ju.edu.jo; 9Department of Internal Medicine, School of Medicine, The University of Jordan, Amman 11942, Jordan; Fbakri@ju.edu.jo; 10Department of Internal Medicine, Jordan University Hospital, Amman 11942, Jordan; 11Infectious Diseases and Vaccine Center, The University of Jordan, Amman 11942, Jordan

**Keywords:** vaccine acceptance, vaccine hesitance, vaccine confidence, anti-vaxxer, conspiracy, COVID-19 vaccine, influenza vaccine

## Abstract

Vaccination could be an effective strategy for slowing the spread of the current coronavirus disease 2019 (COVID-19) pandemic. Vaccine hesitancy could pose a serious problem for COVID-19 prevention, due to the spread of misinformation surrounding the ongoing pandemic. The aim of this study was to assess the attitudes towards the prospective COVID-19 vaccines among the general public in Jordan, Kuwait and other Arab countries. We also aimed to assess the association between COVID-19 vaccine acceptance and conspiracy beliefs. This study used an online survey distributed in December 2020, with items assessing conspiracies regarding COVID-19’s origin and vaccination. Attitudes towards COVID-19 vaccines were assessed using the Vaccine Conspiracy Belief Scale (VCBS), with higher scores indicating a greater belief in vaccine conspiracy. A total of 3414 respondents completed the survey, the majority being residents of Jordan (*n* = 2173, 63.6%), Kuwait (*n* = 771, 22.6%) and Saudi Arabia (*n* = 154, 4.5%). The acceptance rates for COVID-19 and influenza vaccines were 29.4% and 30.9%, respectively. Males, respondents with higher educational levels and those with histories of chronic disease had higher rates of COVID-19 vaccine acceptance. Beliefs that COVID-19 vaccines are intended to inject microchips into recipients and that the vaccines are related to infertility were found in 27.7% and 23.4% of respondents, respectively. Higher VCBS scores were found among females, respondents with lower educational levels and respondents relying on social media platforms as the main source of information. The high rates of vaccine hesitancy in Jordan and Kuwait, among other Arab countries, are alarming. They could hinder the proper control of COVID-19 in the region. The harmful effect of COVID-19 misinformation and conspiracy beliefs was manifested in vaccine hesitancy. This may represent a massive obstacle to the successful control of the pandemic. A reliance on social media as the main source of information about COVID-19 vaccines was associated with vaccine hesitancy. This should alert governments, policy makers and the general public to the importance of vigilant fact checking.

## 1. Introduction

Vaccine hesitancy is the term used to define refusal or reluctance in the acceptance of vaccination despite the availability of vaccination services [1]. The modern endorsement of vaccine hesitancy is a well-known phenomenon, with older roots that have accompanied vaccination since its scientific inception [2,3,4]. This phenomenon has resulted in the resurgence of vaccine-preventable infectious diseases such as measles, poliomyelitis and pertussis [5,6,7].

One year has passed since the official reporting of the first case of infection by the novel severe acute respiratory syndrome coronavirus 2 (SARS-CoV-2) [8,9]. During 2020, the world was devastated by the overwhelming effects of the coronavirus disease 2019 (COVID-19) pandemic, with more than 1.8 million deaths and the exhaustion of healthcare systems, in addition to its negative socio-economic and psychologic impacts [10,11,12,13,14,15].

An essential tool for controlling the ongoing COVID-19 pandemic is the availability of efficacious vaccine(s), which can help in reducing transmission, hospital admissions and the demand on intensive care [16]. Governments supported the pharmaceutical industry and academic community to direct huge efforts towards the development of safe and effective vaccines, among therapeutics and rapid diagnostics [17,18]. To date, more than 100 have been in pre-clinical development, with more than 50 vaccine candidates reaching the clinical development phase [19].

The manufacturing and utility of an efficacious SARS-CoV-2 vaccine face many challenges: selecting an appropriate formulation, the review and approval of an enormous number of potential vaccine candidates, mass manufacturing, and post-marketing surveillance, besides the cost issues and logistics of distribution [20,21,22]. However, a major hindrance to achieving proper vaccination and eventual herd immunity can be vaccine hesitancy among the general public [23]. Vaccine hesitancy is becoming an important obstacle for preventive strategies for combating infectious diseases, and is seen frequently for the prospective SARS-CoV-2 vaccines [24,25]. In addition, a sole dependence on vaccination can result in a worse outcome if other protection strategies are ignored [26].

The basic reproductive number for SARS-CoV-2 was estimated at 2.4–3.4, which would be translated into about 60–72% immune individuals needed to achieve herd immunity [27,28]. Thus, low rates of SARS-CoV-2 vaccine acceptance may pose a serious challenge to controlling this pandemic [23].

Vaccine hesitancy can be attributed to the “3 Cs” model, which points to confidence, complacency and convenience [1]. A lack of confidence in vaccines and providers, complacency towards the need for vaccination, and vaccine inconvenience in terms of unaffordability and costs are the leading factors behind vaccine hesitancy [29,30,31]. Further dissection of vaccine hesitancy reveals the involvement of personal, cultural or religious beliefs [32]. In addition, conspiracy beliefs can lead to vaccine hesitancy through igniting mistrust in governments, healthcare providers and the pharmaceutical industry, besides their known negative impacts on human health behavior [3,33,34,35,36].

For COVID-19, conspiracy beliefs surrounded this pandemic early on [37]. These beliefs revolved around aspects related to the virus being man-made [37]. In addition, such harmful beliefs extended to include notions about the prospective vaccines, such as the accusations of plots to enforce vaccination, which would be used to implant microchips to control humans. Moreover, additional claims that COVID-19 vaccines could lead to infertility, limiting the growth of the human population, gained attention on social media [37,38,39]. Such claims without any evidence circulated on some social media platforms and can have a tremendous negative impact on the general public’s attitude towards the prospective vaccines [37,39,40].

In our previous research on the harmful effects of belief in conspiracy regarding COVID-19 among the students and the general public in Jordan, higher anxiety levels were found to be associated with such beliefs [41,42]. In this study, we aimed to assess the overall acceptance rates for COVID-19 vaccines in Jordan, Kuwait and other Arab countries. Additionally, we aimed to assess the attitude of the general public in these countries towards prospective COVID-19 vaccines. In addition, we aimed to evaluate the harmful effects of belief in conspiracy in relation to prospective COVID-19 vaccination.

## 2. Materials and Methods

### 2.1. Study Design

The data utilized in this cross-sectional study on attitude were collected using an online-based questionnaire, which was conducted between 14 December 2020 (15:00) and 18 December 2020 (21:00), and targeted residents in Jordan aged 16 years and above. Other participants from Arab-speaking countries were invited as well. Potential participant recruitment was performed by advertisement on social media platforms (i.e., Facebook, Instagram and Twitter) and through free messaging services (WhatsApp and Snapchat), starting with contacts of the authors in Jordan and Kuwait.

The eligibility criteria included age more than or equal to 16, current residence in a country where Arabic is an official language, and an ability to read and understand Arabic. To assess the clarity of the survey items in Arabic and to evaluate the average duration for the completion of the survey, a pilot test (*n* = 7) was conducted. The language used to conduct the survey was Arabic (Appendix A). The responses to all items were mandatory, except the item on monthly income, which was restricted to people residing in Jordan.

### 2.2. Ethical Considerations

This study was approved by the Scientific Research Committee at the School of Medicine/University of Jordan (reference number: 5338/2020/67). Participation in the study was voluntary, and an informed consent form was included in the introductory section of the online survey. All collected data were treated with confidentiality.

### 2.3. Survey Items

The final questionnaire comprised four sections with a total of 23 items. The first section on demographics and previous experience with COVID-19 included questions on the following: age, sex, country of residence, educational level, monthly income (for respondents residing in Jordan), history of any chronic disease and previous COVID-19 diagnosis for the respondent or a family member.

The second section comprised eight items that assessed belief in conspiracy about COVID-19’s origin, belief that SARS-CoV-2 was manufactured to force the public to get vaccinated, willingness to get a COVID-19 vaccine if available, willingness to get an influenza vaccine/being vaccinated for influenza, opposition to vaccination in general, belief that the COVID-19 vaccine is a way to implant microchips into people to control them, belief that the COVID-19 vaccine will lead to infertility, and attitude towards potential mandatory COVID-19 vaccination by governments.

The third section assessed the single main source of knowledge about COVID-19 vaccines (allowing the selection of a single main source out of four possible options: television and news releases; social media platforms (Facebook, Twitter and WhatsApp, among others); medical doctors, scientists or scientific journals; or YouTube.

Finally, the fourth section was based on the brief, previously validated Vaccine Conspiracy Beliefs Scale (VCBS), with minor modifications to accommodate questions on prospective COVID-19 vaccines [43]. The participants were asked to indicate how much they agreed or disagreed with each one of seven statements, using a seven-point scale (Appendix A). The scale ranged from “strongly disagree”, which was given a minimum score of 1, to “strongly agree”, which was given a maximum score of 7 [43]. Higher VCBS scores suggest greater belief in vaccine conspiracies.

### 2.4. Statistical Analysis

Statistical analysis was performed using IBM SPSS v22.0 for Windows. Statistical significance was considered for *p* < 0.050. We used the chi-squared (χ^2^) test to analyze associations between categorical variables. For continuous variables (age and VCBS), the mean and standard deviation (SD) were calculated, and analysis with an outcome (e.g., vaccine acceptance and belief in conspiracy) was conducted using the Mann–Whitney *U* test and Kruskal–Wallis (K-W) test. The association between conspiracy beliefs regarding COVID-19’s origin and vaccine acceptance was assessed using multinomial logistic regression with the following covariates: age category, sex, educational level, history of chronic disease, and previous experience of COVID-19 in one’s self or in one’s family.

## 3. Results

### 3.1. Sample Characteristics

The total number of completed surveys was 3414. The countries with at least a hundred respondents were Jordan (*n* = 2173, 63.6%), Kuwait (*n* = 771, 22.6%) and Saudi Arabia (*n* = 154, 4.5%). The characteristics of the participants divided by country of residence are shown in Table 1. Other Arab countries with respondents included Palestine (*n* = 98), Iraq (*n* = 60), United Arab Emirates (*n* = 44), Yemen (*n* = 30), Qatar (*n* = 28), Egypt (*n* = 17), Lebanon (*n* = 9), Oman (*n* = 8), Bahrain (*n* = 6), Tunisia (*n* = 5), Sudan (*n* = 4), Syria (*n* = 3), Somalia (*n* = 2), Algeria (*n* = 1) and Morocco (*n* = 1).

Females predominated in the study sample (*n* = 2299, 67.3%), and the mean age was 31 years (median: 26 years, interquartile range: 21–39 years). The median age among the respondents in Jordan was 24 years, compared to 30 years among the respondents in Kuwait. For the educational level, three-quarters of the study respondents were at undergraduate study level (*n* = 2562).

A history of chronic disease in the study sample was reported by 10.6% of the respondents (*n* = 361), and its distribution was uneven between Jordan and Kuwait (9.4% vs. 13.4%; *p* = 0.002, χ^2^ test). Additionally, previous self/family experience of COVID-19 was less common in Jordan compared to Kuwait (35.9% vs. 50.8%; *p* < 0.001, χ^2^ test). The monthly household income was only reported in Jordan, with the highest category being between 500 Jordanian dinars (JOD) and 1000 JOD (*n* = 826, 38.8%, Table 1).

### 3.2. Rates of Vaccine Acceptance and Related Factors

The overall acceptance of the prospective COVID-19 vaccines among the study sample was 29.4%. Male sex, higher educational level and history of chronic disease were associated with higher rates of COVID-19 vaccine acceptance (Table 2). Stratified by the three countries with most responses, COVID-19 vaccine acceptance was the highest in Saudi Arabia (31.8%), followed by Jordan (28.4%), while the lowest rate was seen in Kuwait (23.6%; *p* = 0.016, χ^2^ test).

For influenza, the overall acceptance rate was 30.9%. Similar to the COVID-19 vaccine acceptance rates, a higher rate of flu vaccine acceptance was seen among males (38.2% vs. 27.4%; *p* < 0.001, χ^2^ test), respondents with higher educational levels (40.4% for postgraduate, 29.5% for undergraduate, and 27.9% for high school or less; *p* < 0.001, χ^2^ test) and respondents with histories of chronic disease (40.7% vs. 29.8%; *p* < 0.001; χ^2^ test, Figure 1). In Jordan, higher monthly income was associated with a higher acceptance rate for COVID-19 and flu vaccines (*p* < 0.001 for both comparisons; χ^2^ test, Figure 1).

### 3.3. Vaccine Acceptance and Conspiracy Beliefs

In the whole study sample, the prevalence the beliefs that COVID-19 is a man-made virus was 59.5% (*n* = 2031). Slightly more than 40% of the study respondents believed that COVID-19 is a man-made disease made to force everyone to get the vaccine (*n* = 1376), and more than one-quarter of the respondents believed that the COVID-19 vaccine is a way to implant microchips into people to control them (*n* = 947, 27.7%). In addition, 23.4% of the respondents stated that COVID-19 vaccines will cause infertility (*n* = 800, Figure 2). Regarding beliefs in vaccine conspiracy, comparisons were made between Jordan and Kuwait, since they had the highest response rates (Table 3). The respondents from Kuwait showed higher beliefs of the following compared to those in Jordan: an artificial origin of the virus, that the disease was man-made to enforce vaccination, microchip implanting and infertility claims (Table 3).

To link the rates of vaccine hesitancy with conspiracy beliefs, we used the items in the second section of the survey and conducted multinomial logistic regression. Belief in conspiracy regarding the origin of the virus and regarding the vaccine was associated with less willingness to get the vaccine (Table 4).

### 3.4. Vaccine Acceptance and VCBS

The reliability of the modified VCBS used in this study to evaluate attitudes towards COVID-19 vaccines was assessed by the Cronbach alpha (0.94). A higher VCBS, implying a higher belief in conspiracy behind COVID-19 vaccines, was seen among females (mean VCBS: 26.3 vs. 24.1 in males; *p* < 0.001, M-W). Additionally, respondents with lower educational levels (undergraduate level or less) had a higher mean VCBS compared to respondents with higher educational levels (postgraduate, mean VCBS: 25.9 vs. 23.4; *p* < 0.001, M-W). In Jordan, participants with monthly income <1000 JOD showed a higher mean VCBS compared to those with a higher income (mean VCBS: 26.5 vs. 22.4; *p* < 0.001, M-W, Figure 3). Analysis by age categories did not show a significant difference (*p* = 0.096, K-W).

Regarding the main source of information about COVID-19 vaccines, lower belief in vaccine conspiracy was seen among those who relied on medical doctors, scientists and scientific journals (mean = 23.9, SD = 11.4), compared to those who relied on TV programs and news releases (mean = 25.7, SD = 10.0), and those who relied on YouTube (mean = 26.1, SD = 10.7). The highest VCBS was seen among those who relied on social media platforms (mean = 27.4, SD = 10.2; *p* < 0.001, K-W, Figure 3).

### 3.5. Sources of Information about COVID-19 Vaccines

The most common source of information about COVID-19 vaccines was reliance on medical doctors, scientists and scientific journals (*n* = 1243, 36.4%). This was followed by TV programs and news releases (*n* = 1083, 31.7%), and social media platforms (*n* = 1029, 30.1%), while YouTube was the main source of information for 59 respondents only (1.7%, Table 5). A dependence on medical doctors, scientists and scientific journals was seen more commonly among males and postgraduate respondents (Table 5). Social media platforms were the main source of information about COVID-19 vaccines among respondents aged 16–21 years, respondents residing in Kuwait, and respondents with educational levels of high school or less.

The likelihood of belief in conspiracy regarding the origin of COVID-19 was higher among respondents who relied on social media platforms (65.8%) compared to those who relied on medical doctors, scientists and scientific journals (49.8%; *p* < 0.001, χ^2^ test). The willingness to get a COVID-19 vaccine was the highest among the respondents who relied on medical doctors, scientists and scientific journals (36.1%), and the lowest among the respondents who relied on social media platforms (22.1%; *p* < 0.001, χ^2^ test). A similar result was seen for the willingness to get flu vaccinations (36.7% vs. 26.0%; *p* < 0.001, χ^2^ test). The ideas of implanting microchips and relation to infertility were seen more frequently among those who relied on social media platforms (33.8% and 27.3%) and less frequently among those who relied on medical doctors, scientists and scientific journals (20.8% and 19.7%; *p* < 0.001 for both comparisons, χ^2^ test).

## 4. Discussion

Vaccination can be considered among the most successful achievements of science; nevertheless, vaccine hesitancy continues to thrive [44]. The ongoing COVID-19 pandemic represents a state of fear, anxiety and uncertainty, which is considered a suitable environment for conspiracies to disseminate in [37,41,42,45,46]. Conspiracy beliefs have infiltrated many aspects of the COVID-19 pandemic, such as the novel virus’ origin and the fallacies about the prospective vaccines [46,47]. Several clinical trials assessing the possibility of achieving efficacious and safe vaccines for COVID-19 have shown promising results [48,49,50]. However, the availability of an effective and safe vaccine per se will not guarantee achieving herd immunity and the control of virus spread [51]. Other factors might play a negative role in control strategies that depend on vaccination, including the duration of protection, cost, and logistics of distribution, among others [22,52,53]. Additionally, vaccine hesitancy appears to be an imminent and serious threat for any hopes of controlling the pandemic [54]. This is especially evident amid the current reports of rapid increases in COVID-19 cases as seen in the United Kingdom [55].

In the current study, we aimed to assess the overall rate of the potential acceptance of prospective COVID-19 vaccines among Arab countries. The highest responses were from Jordan and Kuwait. Most Arab countries have a high burden of COVID-19; hence, vaccination can be a helpful way to slow the spread of infections in these countries. If a COVID-19 vaccine was available, only 29.4% of the respondents in the survey stated that they would get vaccinated. This rate is alarming, since it appears to be among the lowest acceptance rates globally [56]. Recent studies on this subject indicated that the potential acceptance of the prospective COVID-19 vaccines ranged from 57.6% to 68.6% in two studies among adults in the United States [57,58]. A recent global survey involving 19 countries reported a less-than-55% acceptance rate in Russia, and the highest rate of 90% in China [59]. In the aforementioned study, Lazarus et al. found relatively high acceptance rates (>80%) in Asian countries with high trust in governments, in addition to high rates in Brazil, India and South Africa [59]. The acceptance rates reached more than 90% in earlier studies from Indonesia and China [60,61].

A low hesitancy rate (20–25%) was seen in a study that surveyed American and Canadian adults in May 2020 [62]. In Europe, the rates of COVID-19 vaccine hesitancy were 41% in Italy and 26% in France [63,64]. Moreover, vaccine hesitancy in low- and middle-income countries appears to be at a low level [65]. Thus, the vaccine acceptance rate in this study appears to be amongst the lowest in the world to the best of our knowledge [56]. An earlier study from Saudi Arabia reported an acceptance rate of 64.7%; however, this study likely represents a snapshot of an earlier phase of COVID-19 in the country [66]. In this study, respondents from Saudi Arabia showed an acceptance rate of 31.8%; however, the low response rate from the Kingdom precluded the further assessment of this result. Additionally, a lower acceptance rate was seen in Kuwait (23.6%) compared to that in Jordan (28.4%). A higher rate of belief in conspiracy was seen in Kuwait in addition to more dependence on social media platforms to obtain knowledge about the vaccine. This can partly explain the lower acceptance rate for COVID-19 vaccines in Kuwait.

For influenza vaccines, the overall acceptance rate was low. Potential reasons might be related to barriers in accessibility, a fear of adverse reactions, safety concerns and a lack of motivation [67,68]. The results for COVID-19 vaccine acceptance in this study were slightly lower than those for influenza vaccines. It was difficult to evaluate this result from a broader perspective, since earlier studies from the region that investigated flu vaccine hesitancy were limited, focused on health-care workers and reported variable results [69,70]. Variability in influenza vaccine acceptance was also seen in a systematic review by Nguyen et al., with a range of 9–67% [71].

The rate of COVID-19 vaccine acceptance was higher among male respondents compared to females (38.6% vs. 23.9%). This might be correlated with the lesser tendency observed among males to believe in vaccine and virus origin conspiracies, since they mostly relied on medical doctors, scientists and scientific journals, as opposed to females, who relied more on social media platforms. In addition, our previous work showed that females were less likely to perceive the disease as more dangerous, which may result in lower vaccine acceptance due to complacency [42]. Moreover, males residing in Jordan were less likely to believe in conspiracies regarding COVID-19, which is consistent with the results of this study [41,42].

As expected, respondents with higher educational levels had a higher rate of vaccine acceptance, which might be related to their lower tendency to believe in conspiracies. Respondents with histories of chronic diseases were more likely to accept COVID-19 vaccination, which could be related to the higher rates of morbidity and mortality encountered by people with chronic disease [72]. Additionally, a lower monthly income in Jordan was associated with higher rates of vaccine hesitancy.

For the sources of knowledge about the vaccines, respondents who depended on medical doctors, scientists and scientific journals were the least likely to harbor conspiracy beliefs. Thus, the role of such sources in addressing conspiracy issues and providing trusted information should be advocated by the media. Earnshaw et al. reported that doctors were the most trusted source of information about COVID-19, which highlights the significance of their role [73]. On the contrary, social media platforms as sources of information were associated with more doubts and misbeliefs regarding the vaccine. This might be linked to the easier spread of misinformation on social media, including inaccurate information regarding the safety of COVID-19 vaccines [74]. The results were more pronounced among respondents in Kuwait, younger respondents, females, and respondents with lower educational levels. Such a pattern was also seen in our previous work in Jordan [42].

In this study, one of the aims was to link vaccine hesitancy with related conspiracy beliefs. The use of the validated VCBS to achieve this aim can increase the confidence in our results. In this study, higher VCBS scores were associated with significantly higher rates of vaccine hesitancy. This result was independent of other covariates, which delineates one harmful effect of conspiracy belief—namely, vaccine hesitancy. Despite the lack of any evidence to support some claims (e.g., conspiracy plots to use vaccines to implant microchips into humans or as a population-control scheme), around one-quarter of the study sample believed in such misinformation. Such beliefs might seem harmless; however, our results clearly indicate that this may result in a massive negative public health impact, due to its association with vaccine hesitancy.

Conspiracy beliefs—at least in Jordan—appear to worsen over time, since our last survey in April indicated that 47.9% believed in a role of global conspiracy in the origin of COVID-19 [42]. In this study, 58.5% of the respondents in Jordan believed that COVID-19 is a man-made disease, which indicates that conspiracy beliefs are not showing any signs of waning.

A lack of trust in governments, vaccine manufacturers (pharmaceutical companies) and healthcare professionals can lead to the endorsement of conspiracy beliefs [75,76]. The high percentage of adoption of such beliefs shown in this study may be attributed to such mistrust; however, this needs further evaluation to provide conclusive evidence about such a link. Another possible explanation for the adoption of such beliefs could be related to concerns about the perceived safety of the vaccine and the uncertainty regarding the benefits of COVID-19 vaccines, which was cited in research investigating influenza vaccine hesitancy [77,78]. Others would argue that COVID-19 vaccines did not undergo extensive clinical trials and that the long-term side effects are still unknown, and for these reasons, they will abstain from getting the vaccine. In all cases, these issues should be clarified by the scientific community, since the accelerated rate of vaccine development does not appear to come at the expense of the safety and quality of such vaccines, at least in the short term [79].

### Study Limitations

One of the most obvious limitations in this study was the unequal distribution of respondents in different Arab countries, which precluded the generalizability of our results in the region. The exceptions were Jordan and Kuwait, since the large sample sizes in these countries added confidence in our estimates of the vaccine acceptance rates. The high response rate seen in Jordan and Kuwait can be ascribed to the contribution of the authors’ contacts, who mostly resided in these two countries. Further studies are needed from the region to assess the changes in attitude towards COVID-19 vaccines and to evaluate the generalizability of our results.

In addition, the representativeness of the sample in terms of age and sex might have been a caveat, particularly for Kuwait and Saudi Arabia, since the median age for these populations was slightly different compared to for the sample (37 and 32, respectively). The female predominance in the study sample could have been a source of bias, in relation to the higher rates of vaccine hesitancy seen in females. However, this result was in line with our previous papers that showed higher response rates among females [41,42].

## 5. Conclusions

The results of this study show high rates of vaccine hesitancy in Jordan and Kuwait, among other Arab countries. This could pose a serious threat to the preventive measures aimed at controlling COVID-19 spread in the region. The association of conspiracy beliefs regarding the prospective vaccines and the origin of the virus with vaccine hesitancy should raise a red flag and alert policy makers, governments and different media platforms to the serious harmful effects of spreading misinformation. The consequences for public health could be imminent if these conspiracy beliefs are not challenged with fact checking and evidence-based scientific information.

## Figures and Tables

**Figure 1 vaccines-09-00042-f001:**
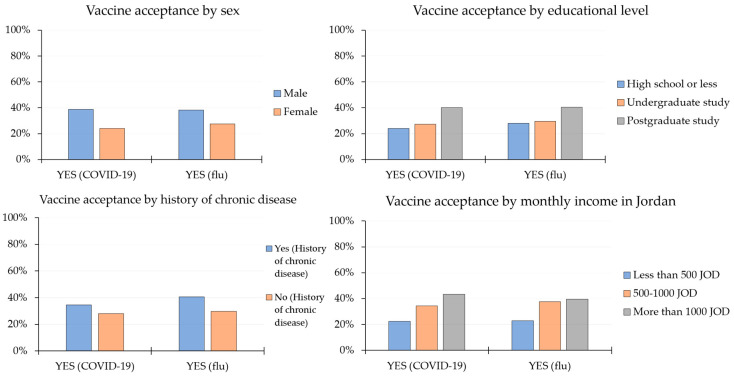
Vaccine acceptance rates stratified by sex, educational level, history of chronic disease and monthly income (for respondents in Jordan only). Analysis was conducted for COVID-19 prospective vaccines and influenza (flu) vaccines. COVID-19: Coronavirus disease 2019; flu: Influenza; JOD: Jordanian dinars.

**Figure 2 vaccines-09-00042-f002:**
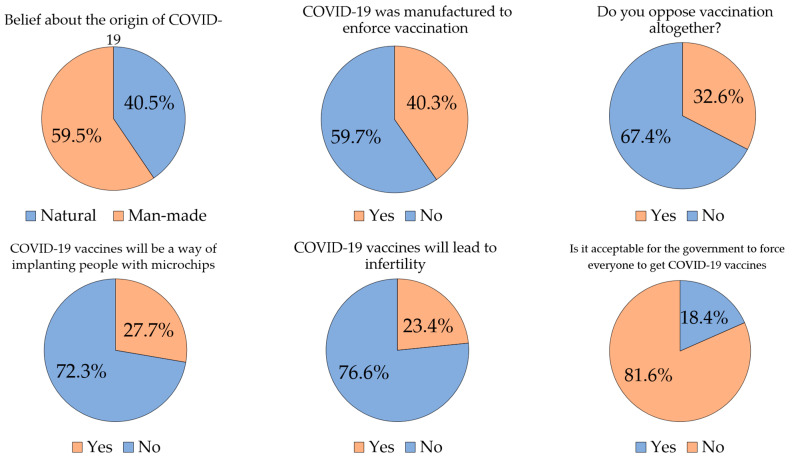
Attitudes to COVID-19 vaccines in relation to conspiracy beliefs and towards governmental enforcement of the vaccines. COVID-19: Coronavirus disease 2019.

**Figure 3 vaccines-09-00042-f003:**
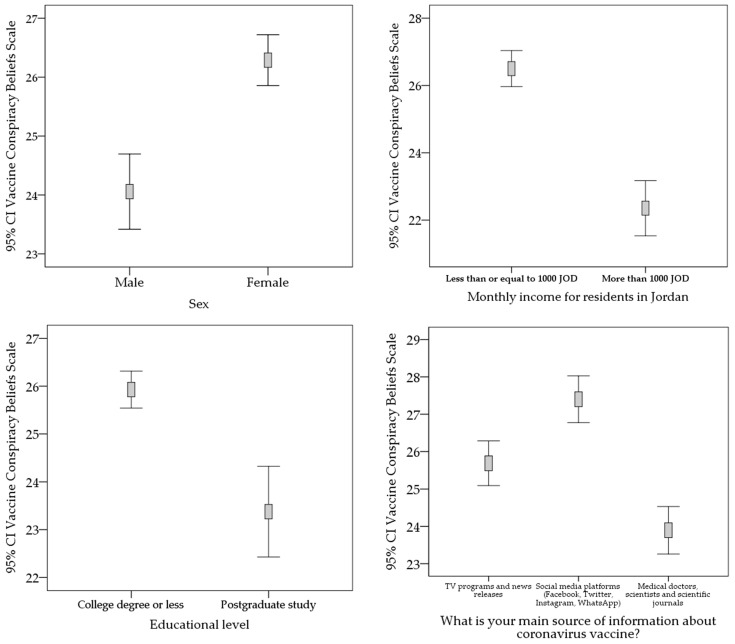
COVID-19 Vaccine Conspiracy Beliefs Scale correlation with sex, educational level and monthly income. CI: confidence interval of the mean; JOD: Jordanian dinar; monthly income was assessed only among respondents in Jordan.

**Table 1 vaccines-09-00042-t001:** Characteristics of the study sample stratified by country of residence.

Country of Residence		Jordan	Kuwait	Saudi Arabia	Others ^5^
		n ^4^ (%)	n (%)	n (%)	n (%)
Respondents		2173 (63.6)	771 (22.6)	154 (4.5)	316 (9.3)
Age category based on quartiles ^1^	16–21 years	755 (34.7)	156 (20.2)	10 (6.5)	68 (21.5)
	22–26 years	480 (22.1)	163 (21.1)	21 (13.6)	120 (38.0)
	27–39 years	419 (19.3)	254 (32.9)	49 (31.8)	76 (24.1)
	40 years or older	519 (23.9)	198 (25.7)	74 (48.1)	52 (16.5)
Sex	Male	665 (30.6)	278 (36.1)	36 (23.4)	136 (43.0)
	Female	1508 (69.4)	493 (63.9)	118 (76.6)	180 (57.0)
Monthly income for residents in Jordan ^2^	Less than 500 JOD	688 (32.3)	-	-	-
	500–1000 JOD	826 (38.8)	-	-	-
	More than 1000 JOD	616 (28.9)	-	-	-
Educational level ^3^	High school or less	173 (8.0)	137 (17.8)	22 (14.3)	27 (8.5)
	Undergraduate	1646 (75.7)	560 (72.6)	111 (72.1)	245 (77.5)
	Postgraduate	354 (16.3)	74 (9.6)	21 (13.6)	44 (13.9)
History of chronic disease	Yes	205 (9.4)	103 (13.4)	21 (13.6)	32 (10.1)
	No	1968 (90.6)	668 (86.6)	133 (86.4)	284 (89.9)
Experience of COVID-19 in self or family	Yes	781 (35.9)	392 (50.8)	60 (39.0)	122 (38.6)
	No	1392 (64.1)	379 (49.2)	94 (61.0)	194 (61.4)

^1^ Age category based on quartiles: The quartiles were determined based on age distribution and interquartile range for the whole study sample. ^2^ Monthly income for residents in Jordan: Only participants residing in Jordan were asked about monthly income in Jordanian dinars (JOD); the total numbers do not add up to 2173, since a response to this item was optional and was only directed to participants in Jordan. ^3^ Educational level: Undergraduate study includes diploma or Bachelor of Science study, while postgraduate study includes Master of Science or Doctor of Philosophy study. ^4^ n: Number. ^5^ Others: Other countries with respondents for the survey; included Palestine = 98, Iraq = 60, United Arab Emirates = 44, Yemen = 30, Qatar = 28, Egypt = 17, Lebanon = 9, Oman = 8, Bahrain = 6, Tunisia = 5, Sudan = 4, Syria = 3, Somalia = 2, Algeria = 1 and Morocco = 1.

**Table 2 vaccines-09-00042-t002:** Possible factors affecting acceptance of COVID-19 vaccines within the study sample.

Variable		Will You Get COVID-19 Vaccine When Available?		*p*-Value ^4^
		Yes	No	
		n ^3^ (%)	n (%)	
Age (mean, SD) ^1^		31.4 (13.4)	30.7 (11.8)	0.908
Sex	Male	430 (38.6)	685 (61.4)	<0.001
	Female	550 (23.9)	1749 (76.1)	
Educational level ^2^	High school or less	86 (24.0)	273 (76.0)	<0.001
	Undergraduate	696 (27.2)	1866 (72.8)	
	Postgraduate	198 (40.2)	295 (59.8)	
History of chronic disease	Yes	125 (34.6)	236 (65.4)	0.009
	No	855 (28.0)	2198 (72.0)	
Experience of COVID-19 in self or family	Yes	384 (28.3)	971 (71.7)	0.701
	No	596 (28.9)	1463 (71.1)	

^1^ SD: Standard deviation. ^2^ Educational level: Undergraduate study includes diploma or Bachelor of Science study, while postgraduate study includes Master of Science or Doctor of Philosophy study. ^3^ n: Number. ^4^
*p*-value: For age comparisons, we used Mann–Whitney *U* tests, and for categorical variables, we used chi-squared tests.

**Table 3 vaccines-09-00042-t003:** Vaccine conspiracy beliefs and attitude items stratified by countries of residence.

Country of Residence		Jordan	Kuwait	*p*-Value ^3^	Saudi Arabia	Others ^4^
Conspiracy Belief and Attitude Items		n ^2^ (%)	n (%)	(Jordan vs. Kuwait)	n (%)	n (%)
What is your belief about the origin of the current coronavirus in humans?	Natural	902 (41.5)	272 (35.3)	0.002	53 (34.4)	156 (49.4)
	Man-made	1271 (58.5)	499 (64.7)		101 (65.6)	160 (50.6)
Do you think the current coronavirus was man-made to force everyone to get vaccinated?	Yes	833 (38.3)	373 (48.4)	<0.001	73 (47.4)	97 (30.7)
	No	1340 (61.7)	398 (51.6)		81 (52.6)	219 (69.3)
Will you get the coronavirus vaccine when available?	Yes	618 (28.4)	182 (23.6)	0.010	49 (31.8)	131 (41.5)
	No	1555 (71.6)	589 (76.4)		105 (68.2)	185 (58.5)
Have you had or are you going to have the influenza vaccine?	Yes	643 (29.6)	246 (31.9)	0.229	47 (30.5)	120 (38.0)
	No	1530 (70.4)	525 (68.1)		107 (69.5)	196 (62.0)
Do you oppose vaccination altogether?	Yes	706 (32.5)	279 (36.2)	0.062	58 (37.7)	69 (21.8)
	No	1467 (67.5)	492 (63.8)		96 (62.3)	247 (78.2)
Do you think that coronavirus vaccine will be a way of implanting people with microchips to control humans?	Yes	604 (27.8)	247 (32.0)	0.026	34 (22.1)	62 (19.6)
	No	1569 (72.2)	524 (68.0)		120 (77.9)	254 (80.4)
COVID-19 vaccines will lead to infertility ^1^	Yes	504 (23.2)	212 (27.5)	0.017	32 (20.8)	52 (16.5)
	No	1669 (76.8)	559 (72.5)		122 (79.2)	264 (83.5)
Do you think it is acceptable for the government to force everyone to get coronavirus vaccine?	Yes	410 (18.9)	126 (16.3)	0.118	28 (18.2)	63 (19.9)
	No	1763 (81.1)	645 (83.7)		126 (81.8)	253 (80.1)

^1^ COVID-19: Coronavirus disease 2019. ^2^ n: Number. ^3^
*p*-value: Calculated using chi-squared test. ^4^ Others: Other countries with respondents for the survey; included Palestine = 98, Iraq = 60, United Arab Emirates = 44, Yemen = 30, Qatar = 28, Egypt = 17, Lebanon = 9, Oman = 8, Bahrain = 6, Tunisia = 5, Sudan = 4, Syria = 3, Somalia = 2, Algeria = 1 and Morocco = 1.

**Table 4 vaccines-09-00042-t004:** Results of multinomial regression analysis of factors associated with vaccine acceptance.

Factor	Odds Ratio (95% CI) ^2^	*p*-Value
COVID-19 origin (natural vs. man-made) ^1^	0.47 (0.38–0.57)	<0.001
COVID-19 is man-made to force people to get the vaccine? (yes vs. no)	1.89 (1.46–2.43)	<0.001
COVID-19 vaccine will be used to implant microchips to humans? (yes vs. no)	2.39 (1.72–3.30)	<0.001
COVID-19 vaccine causes infertility? (yes vs. no)	2.73 (1.90–3.93)	<0.001
Are you against vaccination in general? (yes vs. no)	9.26 (6.60–12.99)	<0.001
**Covariates**		
Age category	1.01 (0.93–1.11)	0.762
Sex	1.54 (1.28–1.85)	<0.001
Country of residence	0.94 (0.90–0.98)	0.005
Educational level	0.78 (0.64–0.94)	0.010
History of chronic disease	1.55 (1.15–2.09)	0.004
Self or family experience of COVID-19	0.88 (0.74–1.06)	0.178

^1^ COVID-19: Coronavirus disease 2019. ^2^ CI: Confidence interval.

**Table 5 vaccines-09-00042-t005:** The main sources of information regarding COVID-19 vaccines stratified by respondent demographics.

Characteristic	Main Source of Information about COVID-19 Vaccines	TV Programs and News Releases	Social Media Platforms	Medical Doctors, Scientists and Scientific Journals	YouTube	*p*-Value ^5^
		n ^4^ (%)	n (%)	n (%)	n (%)	
Age category based on quartiles ^1^	16–21 years	307 (31.0)	337 (34.1)	320 (32.4)	25 (2.5)	<0.001
	22–26 years	199 (25.4)	269 (34.3)	304 (38.8)	12 (1.5)	
	27–39 years	262 (32.8)	222 (27.8)	299 (37.5)	15 (1.9)	
	40 years or older	315 (37.4)	201 (23.8)	320 (38.0)	7 (0.8)	
Sex	Male	303 (27.2)	310 (27.8)	481 (43.1)	21 (1.9)	<0.001
	Female	780 (27.2)	719 (31.3)	762 (33.1)	38 (1.7)	
Country of residence ^2^	Jordan	756 (34.8)	581 (26.7)	804 (37.0)	32 (1.5)	<0.001
	Kuwait	193 (25.0)	304 (39.4)	257 (33.3)	17 (2.2)	
	Saudi Arabia	53 (34.4)	45 (29.2)	53 (34.4)	3 (1.9)	
	Others	81 (25.6)	99 (31.3)	129 (40.8)	7 (2.2)	
Educational level ^3^	High school or less	108 (30.1)	148 (41.2)	89 (24.8)	14 (3.9)	<0.001
	Undergraduate	839 (32.7)	809 (31.6)	875 (34.2)	39 (1.5)	
	Postgraduate	136 (27.6)	72 (14.6)	279 (56.6)	6 (1.2)	

^1^ Age category based on quartiles: The quartiles were determined based on age distribution and interquartile range for the whole study sample. ^2^ Others: Other countries with respondents for the survey; included Palestine = 98, Iraq = 60, United Arab Emirates = 44, Yemen = 30, Qatar = 28, Egypt = 17, Lebanon = 9, Oman = 8, Bahrain = 6, Tunisia = 5, Sudan = 4, Syria = 3, Somalia = 2, Algeria = 1 and Morocco = 1. ^3^ Educational level: Undergraduate study includes diploma or Bachelor of Science study, while postgraduate study includes Master of Science or Doctor of Philosophy study. ^4^ n: Number. ^5^
*p*-value: Calculated using chi-squared test.

## Data Availability

The data presented in this study are available on request from the corresponding author (M.S.). The data are not publicly available due to privacy concerns since we obtained few responses from several countries other than Jordan, Kuwait and Saudi Arabia.

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
