# Peer review of "High Rates of COVID-19 Vaccine Hesitancy and Its Association with Conspiracy Beliefs: A Study in Jordan and Kuwait among Other Arab Countries"

_vaccines, 2021, doi:10.3390/vaccines9010042_

Round 1

Reviewer 1 Report

This article focuses on a very interesting topic that many societies have to deal with nowadays, vaccine hesitancy. Understanding the underlining reasons of this phenomenon may be extremely helpful for the medical society and countries' governments to tackle the problem that rises and becomes an obstacle in the fight towards the pandemic of COVID-19.

The authors, through the questionnaire and the results that come up by it,  provide sufficient information which can be used by the investigated countries in order to organize campaigns and initiatives that will diminish any reluctance and lead to the acceptance of being vaccinated.

However, there are some minor grammar and spelling errors throughout the manuscript. Thus, I would recommend to have a native English speaker check it.

Author Response

Response: We would like to thank the reviewer for his/her comments. For the minor grammar and spelling errors, we conducted a language reviewing and editing of the manuscript by an expert who was acknowledged in the acknowledgment section.

The following changes were made following the language editing process:

Page 2, line 64; lines 74-75; lines 75-78; line 80

Page 3, lines 141-142; lines 144-145

Page 4, line 176; line 183; line 186; line 189

Page 6, line 226

Page 8, lines 260-261

Page 9, lines 267, 270, 271 and 273

Page 11, lines 338-343; lines 363-366; lines 372-373

Page 12, lines 377-380; lines 419-423

Reviewer 2 Report

This research article by Sallam M. et al. clarified association between vaccine conspiracy beliefs, gender, incomes, educational level, history of chronic diseases and the attitude towards the prospective COVID-19 vaccines among the general public in Jordan, Kuwait and other Arab countries. The authors compared their current study and previous publications, mentioned the limitation of the study they performed and outlined factors that affected acceptance of the COVID-19 vaccines in Arab countries and the other countries. There are some papers assessing the similar topics in other areas, but in my opinion, the data in this article provides meaningful information and opinions on this field timely and it fits well to the scope of this journal. Hopefully, the authors could get more respondents, but I will recommend the publication of the manuscript in present form.

My additional minor comment is length of abstract. Authors should describe abstract concisely (Vaccines: The abstract could be a total of about 200 words maximum).  

Author Response

Response: We would like to thank the reviewer for his/her comments.

For the length of the abstract, we made several changes in order to make it shorter and more concise. Please refer to the revised highlighted manuscript (Pages: 1-2, Lines: 25-52).

Reviewer 3 Report

Estimated Authors,

Estimated Editors,

thank you for the opportunity to review this very interesting paper on the vaccine hesitancy and its association with conspiracy beliefs in Arab countries.

The paper is well written since its inception to its discussion.

Not only I've no significant suggestions for improvements, but I urge for a rapid publication because of its meaning and the potential public health implications in Middle East.

Author Response

Response: We are deeply grateful, and we would like to thank the reviewer for his/her comments.